

# Antimicrobial activities of Diltiazem Hydrochloride: drug repurposing approach

Omar K. Alduaij[1],*, Rageh K. Hussein[1], Sharif Abu Alrub[1] and Sabry A. H. Zidan[2],*

[1] Department of Physics, College of Science, Imam Mohammad Ibn Saud Islamic University (IMSIU), Riyadh, Saudi Arabia
[2] Department of Pharmacognosy, Faculty of Pharmacy, Al-Azhar University, Assiut-Branch, Assiut, Egypt
* These authors contributed equally to this work.

## ABSTRACT

**Background:** The growing concern of antibiotic-resistant microbial strains worldwide has prompted the need for alternative methods to combat microbial resistance. Biofilm formation poses a significant challenge to antibiotic efficiency due to the difficulty of penetrating antibiotics through the sticky microbial aggregates. Drug repurposing is an innovative technique that aims to expand the use of non-antibiotic medications to address this issue. The primary objective of this study was to evaluate the antimicrobial properties of Diltiazem HCl, a 1,5-benzothiazepine $Ca_2^+$ channel blocker commonly used as an antihypertensive agent, against four pathogenic bacteria and three pathogenic yeasts, as well as its antiviral activity against the Coxsackie B4 virus (CoxB4).

**Methods:** To assess the antifungal and antibacterial activities of Diltiazem HCl, the well diffusion method was employed, while crystal violet staining was used to determine the anti-biofilm activity. The MTT (3-(4,5-dimethylthiazol-2-yl)-2,5-diphenyltetrazolium bromide) colorimetric assay was utilized to evaluate the antiviral activity of Diltiazem HCl against the CoxB4 virus.

**Results:** This study revealed that Diltiazem HCl exhibited noticeable antimicrobial properties against Gram-positive bacteria, demonstrating the highest inhibition of *Staphylococcus epidermidis*, followed by *Staphylococcus aureus*. It effectively reduced the formation of biofilms by 95.1% and 90.7% for *S. epidermidis*, and *S. aureus*, respectively. Additionally, the antiviral activity of Diltiazem HCl was found to be potent against the CoxB4 virus, with an $IC_{50}$ of 35.8 ± 0.54 µg mL$^{-1}$ compared to the reference antiviral Acyclovir ($IC_{50}$ 42.71 ± 0.43 µg mL$^{-1}$).

**Conclusion:** This study suggests that Diltiazem HCl, in addition to its antihypertensive effect, may also be a potential treatment option for infections caused by Gram-positive bacteria and the CoxB4 viruses, providing an additional off-target effect for Diltiazem HCl.

Corresponding author
Sabry A. H. Zidan,
sabryzidan@azhar.edu.eg

## INTRODUCTION

Infectious diseases caused by pathogenic microorganisms including bacteria, viruses, and fungi pose a significant threat to human health (*Soni, Sinha & Pandey, 2024*). The emergence of multidrug-resistant pathogens has become a major concern worldwide, presenting a challenge to the pharmaceutical industry for low-cost development of effective medications against these microorganisms (*Mubarack et al., 2012*). In particular, pathogens such as *S. aureus* and *S. epidermidis* have developed the ability to form biofilms, which can lower the effectiveness of antibiotics because of their resistance to antibiotics penetration (*Mubarack et al., 2012*).

Furthermore, viral resistance has emerged as a problem when treating viral diseases, particularly in immunocompromised patients (*Karimi et al., 2013*). The CoxB4 virus, for example, is a significant human pathogen that can cause mild symptoms such as fever, rash, and upper respiratory illness (*Taylor et al., 2020*). In rare cases, it can also lead to more severe conditions such as meningoencephalitis, myocarditis, and moderate-to-severe pancreatitis (*Taylor et al., 2020*; *Huber & Ramsingh, 2004*). Although acute pancreatitis is often mild, it can lead to systemic inflammation and even death in up to 30% of patients (*Hochman, Louie & Bailey, 2006*). Therefore, it is crucial to conduct ongoing research to identify low-cost and low-side-effect antimicrobial agents that can effectively treat these damaging microbes.

Drug repurposing has potential method to expand the application of existing drugs that are in use and have passed all clinical trials, which saves time, effort, and cost for pharmaceutical firms in the production of new drugs (*Weth et al., 2024*). Diltiazem, a voltage-gated $Ca_2^+$ channel antagonist, was the first 1,5-benzothiazepine derivative described to treat angina pectoris, hypertension, supraventricular tachyarrhythmia, and other related cardiac disorders (*Bariwal et al., 2008*). The hydrochloride salt of Diltiazem (Diltiazem HCl) has a vasodilation effect on the hepatic superior mesenteric and femoral arteries (*Adaramoye et al., 2009*). Recent repurposing of Diltiazem for treating different illnesses has shown promising outputs, such as those for Alzheimer's disease (*Alluri et al., 2023*), inhibition of influenza infection (*Pizzorno et al., 2019*), radioprotection (*Kaur et al., 2023*), and antitumor effects (*Ribeiro et al., 2023*). However, other potential targets, such as its antibacterial, antifungal, and anti-CoxB4 activities, remain to be explored. Accordingly, the main goal of the present study was to investigate, for the first time, the antibacterial activity of Diltiazem HCl against four pathogenic bacteria (*Escherichia coli* ATCC 8739, *S. epidermidis* ATCC 12228, *S. aureus* ATCC 6538, and *Serratia marcescens* AUMC B-58), and three pathogenic yeasts (*Candida albicans* AUMC 13415, *Candida glabrata* AUMC 13412, and *Candida krusei* AUMC 13420), as well as its antiviral activity against the CoxB4 virus, and to evaluate its ability to inhibit biofilm formation by the affected bacteria, aiming to expand its off-target application as an antimicrobial agent.

## MATERIALS AND METHODS

### Antimicrobial activity of Diltiazem HCl

The antimicrobial potential of Diltiazem HCl (Sigma-Aldrich, St. Louis, MO, USA) was performed against three pathogenic yeasts (*Candida albicans* AUMC 13415, *Candida*

*glabrata* AUMC 13412, and *Candida krusei* AUMC 13420), and four pathogenic bacterial strains (*Escherichia coli* ATCC 8739, *S. epidermidis* ATCC 12228, *S. aureus* ATCC 6538, and *Serratia marcescens* AUMC B-58), using the well diffusion method as described by *Valgas et al. (2007)*. Diltiazem HCl was used at concentrations of 50, 25, 12.5, and 6.32 mg mL$^{-1}$. Fluconazole and Chloramphenicol are among the most widely used drugs to treat fungal and bacterial infections, respectively (*Russell, 2004*; *Abuhammour & Habte-Gabr, 2001*), therefore, they were used as reference drugs at a concentration of 12.5 mg mL$^{-1}$. The examined organisms were cultivated in 24-hour-old cultures. Using sterile cotton swabs, the bacterial and fungal suspensions ($10^8$ CFU/mL = 0.5 McFarland standard solution) were equally spread on sterile Petri plates filled with either nutrient agar for bacteria or Sabouraud's dextrose agar for *Candida* species. Five wells (6 mm diameter holes cut in solid medium) were filled with 50 µL of the substance under test. The plates were incubated for 24 h at 37 °C to allow the bacteria and *Candida* species to grow. Following incubation, the suppressions of fungal and bacterial growth were measured in millimeters. The minimum bactericidal concentration (MBC) was determined. All tests were carried out in triplicate.

## Microtitre plate assay for biofilm quantification

The effect of Diltiazem HCl on biofilm formation was evaluated as described by *Mohanta et al. (2020)*. Briefly, using a 96-well polystyrene plate, 300 µL of fresh tryptone soy yeast broth inoculated with $10^6$ CFU mL$^{-1}$ was aliquoted into each well and cultured in the presence of 75%, 50%, and 25% of MBC which was determined in the previous step. Wells with bacteria but no drugs were used as a control, while those containing medium without bacteria were used as a blank. The plates were then incubated for 48 h at 37 °C. After the incubation period, the supernatant was disposed of and every well was meticulously cleaned using sterile distilled water to eliminate any floating cells. After 30 min of air drying, the biofilm formed was stained for 15 min at 24 °C using an aqueous solution of crystal violet (0.1%). Finally, 250 µL of 95% ethanol was added to each well to solubilize the dye. The plate was then incubated for 15 min and the optical density (OD) was recorded at 570 nm using a microplate reader (SN 18052112; USA). To calculate the biofilm inhibition ability of Diltiazem HCl, the following formula was used:

$$\text{Biofilm dispersion assay} = 1 - \frac{\text{OD of cells treated with Diltiazem HCl}}{(\text{OD of the non-treated cells} - \text{OD of treated control})} \times 100$$

## Antiviral activity
### Virus, strains, and cell culture conditions

The CoxB4 virus and the normal Vero CCL-81 cell line were obtained from the Microbiology Department, Faculty of Medicine, Al-Azhar University, Cairo, Egypt. The normal Vero cells were cultivated in RPMI 1640 medium (Gibco, Tunisia) enriched with fetal bovine serum (10% v/v), L-glutamine (2 mM), penicillin (100 U mL$^{-1}$), and streptomycin (100 µg mL$^{-1}$), and then incubated at 37 °C in a humidified atmosphere with 5% $CO_2$ in an incubator (Jouan, France).

### Determination of cytotoxicity on VERO cells

The maximum non-toxic concentration (MNTC) of Diltiazem HCl on VERO cells was determined using the MTT colorimetric assay as described by *Zidan et al. (2020)*. Briefly, different concentrations of the drug were prepared by double-fold dilutions. After the formation of a confluent sheet of VERO cells, the growth medium, Dulbecco's modified Eagle medium (DMEM), was poured out from the 96-well microtiter plates. The cell monolayer was washed twice with rinse Dulbecco's phosphate buffered saline medium, and 0.1 mL of each dilution was added separately in different wells, leaving three wells as controls. The plate was incubated at 37 °C for up to 2 days. The cells were examined for any physical signs of toxicity such as partial or complete monolayer loss, rounding, shrinkage, or cell granulation. MTT solution (5 mg mL$^{-1}$ in PBS) was prepared (Bio Basic Canada. Inc). Each well received 20 µL of MTT solution before shaking at 150 rpm for 5 min to thoroughly mix the MTT into the medium. The plate was incubated at (37 °C and 5% $CO_2$) for 1.5 h in a $CO_2$-incubator (Jouan, France) to allow MTT to be metabolized, and the medium was removed. The formed formazan crystals (MTT metabolic product) were re-suspended in 200 mL of DMSO and shaken at 150 rpm for 5 min to mix well. At 560 nm, the OD was calculated, and at 620 nm the background was removed. The OD and cell count were closely related. The non-lethal concentration of the drug was determined, and the test was performed in triplicate. The cell viability (%) was calculated using the following equation:

$$\text{Cell viability } (\%) = \frac{(\text{Mean OD of drug} - \text{treated cells})}{(\text{Mean OD of untreated control cells})} \times 100$$

### MTT assay protocol

Antiviral activity was assessed using the MTT assay as described by *Zidan et al. (2016)*. With 10,000 cells overlaid in 200 µL of DMEM per well in a 96-well plate, different dilutions of the drug at its nonlethal dose, and the virus suspension were incubated at equal volumes (1:1 v/v) for 1 h. The viral/sample suspension (100 µL) was added, and the mixture was shaken at 150 rpm for 5 min. Three wells were left empty as blank controls. To provide sufficient time for the virus to function, the cells were then allowed to attach to the wells at 37 °C with 5% $CO_2$ overnight in a $CO_2$-incubator (Jouan, France). Two milliliters of MTT solution (5 mg mL$^{-1}$) were prepared in PBS for each 96-well plate. The MTT solution (20 µL) was added to each well. The plate was placed on a shaking table at 150 rpm for 5 min to mix the MTT into the medium systematically. The plate was then incubated at (37 °C and 5% $CO_2$) for 1.5 h to allow the MTT to be digested before the medium was removed. Formazan crystals were re-suspended in 200 µL DMSO and shaken for 5 min at 150 rpm on a shaking table to mix thoroughly. The OD was measured at 560 nm, and the background was subtracted at 620 nm. The OD was proportional to the number of cells. The following equation was used to calculate the antiviral activity of Diltiazem HCl:

$$\text{Antiviral activity (\%)} = \frac{(\text{OD of treated cells} - \text{OD of virus control})}{(\text{OD of control cells} - \text{OD of virus control})} \times 100$$

The experiment was carried out in triplicate. Acyclovir (Sigma-Aldrich, St. Louis, MO, USA) is a common antiviral medication (*Mehta, 2013*). Therefore, it was used as the reference antiviral agent. The $IC_{50}$, which indicates the concentration of the samples required to inactivate 50% of the virus particles compared to the untreated control, was determined from the dose-response curves. The selectivity index (SI) was calculated by dividing the half-maximal inhibitory concentration of normal Vero cells ($CC_{50}$) by the specific $IC_{50}$ of the virally infected cells.

## Statistical calculation

Statistical analyses were carried out using Microsoft Excel (XLSTAT 2018.3.16, Florida, USA) and GraphPad Prism version 7 software.

## RESULTS AND DISCUSSION

### Antimicrobial activity of Diltiazem HCl

Unfortunately, many traditional antibiotics have failed to treat patients when used over extended periods or improperly, due to the antimicrobial resistance developed by several pathogenic microorganisms (*Ramadan et al., 2024*). Therefore, the search for alternative antibiotics rather than conventional antibiotics is urgently needed (*Ramadan et al., 2023*). Drug repurposing is a productive approach for the discovery of new therapeutic uses for already-available drugs that cut down the time and cost required to develop safe drugs (*Singh et al., 2020*). In this study, the antimicrobial activity of the commonly prescribed antihypertensive drug, Diltiazem HCl (*Stepanovs et al., 2016*) was preliminarily evaluated at concentrations of 50, 25, 12.5, 6.25, and 3.12 mg mL$^{-1}$ against three fungal pathogenic strains of *Candida* sp and four pathogenic bacterial strains (two Gram-positive and two Gram-negative) using the agar well diffusion method. The zones of inhibition were measured (Fig. 1), and the data is presented in 'Table 1'. The results of the current study revealed that Diltiazem HCl efficiently suppressed the growth of tested Gram-positive bacteria. The highest zone of inhibition was recorded against *S. epidermidis* (20 ± 3 mm) at a concentration of 50 mg mL$^{-1}$. The MIC for Diltiazem HCl was 6.25 mg mL$^{-1}$ when employed against *S. epidermidis* with an inhibition zone (10 ± 2.0 mm). The inhibition zone for *S. aureus* was 12 ± 2 mm, and the minimum inhibitory concentration (MIC) was 50 mg mL$^{-1}$ (Table 1, Fig. 1), compared to the Chloramphenicol antibacterial standard. Diltiazem HCl showed a weak antifungal activity against *C. glabrata* with inhibition zone 12 ± 1.0 mm at 50 mg mL$^-$ compared to the antifungal standard Fluconazole. However, the yeasts *C. albicans* and *C. krusei* as well as the tested Gram-negative bacteria (*S. marcescens* and *E. coli*) were unaffected by Diltiazem HCl (Table 1, Fig. 1).

A variety of "non-antibiotics" which are involved in the management of diseases of non-infectious etiology, have been reported to exhibit antimicrobial activity against bacteria and fungi (*Glajzner, Bernat & Jasińska-Stroschein, 2024*; *Kruszewska, Zareba & Tyski, 2008*). 1,5-Benzothiazepine moiety itself has no antimicrobial activity; however,

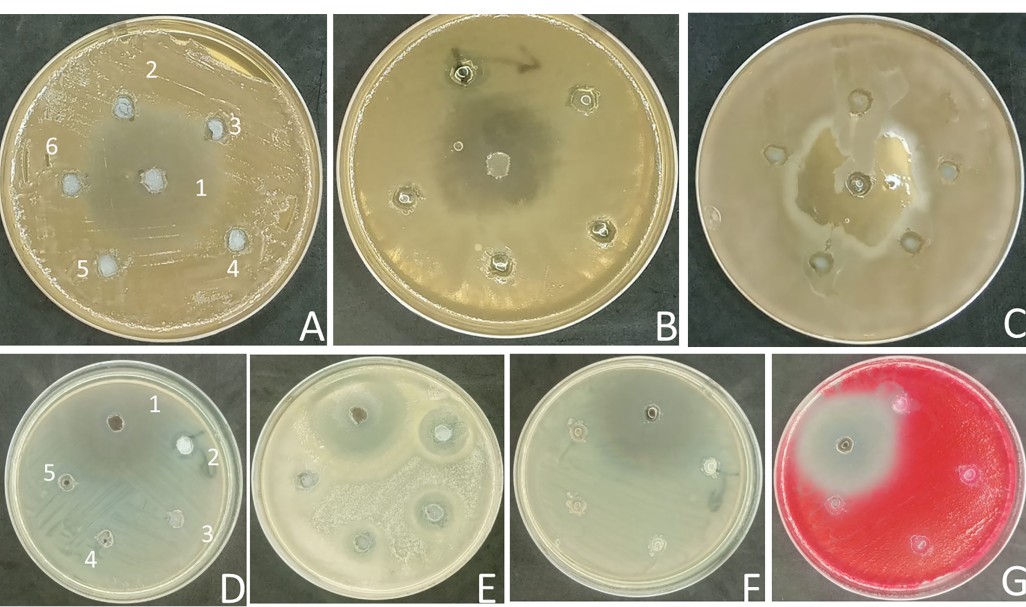

**Figure 1 Antimicrobial activity of Diltiazem HCl (zone of inhibition in mm).** (A) *C. albicans.* (B) *C. glabrata.* (C) *C. krusei.* (D) *S. aureus.* (E) *S. epidermidis.* (F) *E. coli.* (G) *S. marcescens.* (1 = standard, 2 = 50 µg mL$^{-1}$, 3 = 25 µg mL$^{-1}$, 4 = 12.5 µg mL$^{-1}$, 5 = 6.25 µg mL$^{-1}$, and 6 = 3.1 µg mL$^{-1}$).

**Table 1 The antibacterial and antifungal activity of Diltiazem HCl compared to the Fluconazole and Chloramphenicol standards.**

| Tested organisms | | Diltiazem HCl mg mL$^{-1}$ | | | | | Fluconazole | Chloramphenicol |
|---|---|---|---|---|---|---|---|---|
| | | 50 | 25 | 12.5 | 6.25 | 3.1 | 12.5 | 12.5 |
| *Fungi* | *C. albicans* | 0 | 0 | 0 | 0 | 0 | 30 ± 2 | - |
| | *C. glabrata* | 12 ± 1 | 0 | 0 | 0 | 0 | 37 ± 2 | - |
| | *C. krusei* | 0 | 0 | 0 | 0 | 0 | 28 ± 2 | - |
| *Bacteria* | *E. coli* | 0 | 0 | 0 | 0 | 0 | - | 35 ± 2 |
| | *S. aureus* | 12 ± 2 | 0 | 0 | 0 | 0 | - | 33 ± 2 |
| | *S. epidermidis* | 20 ± 3 | 16 ± 2 | 13 ± 3 | 10 ± 2 | 0 | - | 31 ± 2 |
| | *S. marcescens* | 0 | 0 | 0 | 0 | 0 | - | 35 ± 2 |

**Note:**
The activity was measured as zone of inhibition in mm (mean ± SD, $n$ = 3).

many derivatives have been reported to have variable antibacterial and antifungal activities according to the different substitutions of the main nucleus (*Pant, Avinash & Yadav, 2014*; *Dandia et al., 2010*; *Saini, Joshi & Joshi, 2008*). Calcium channel blockers, such as Diltiazem HCl, at their therapeutic dosage, reduce the elevated blood pressure of hypertensive patients but do not affect the blood pressure of normotensive individuals because they have little effect on the peripheral vasculature (*Godfraind, 2017*). So, they can be used safely as anti-infectious agents for both hypertensive and non-hypertensive patients. The observed results demonstrated the noticeable efficacy of Diltiazem HCl, a

**Table 2 Antibiofilm activity (%) of Diltiazem HCl and specific biofilm formation (SBF) against *S. aureus* and *S. epidermidis*.**

| Sample | *S. aureus* | | *S. epidermidis* | |
|---|---|---|---|---|
| | SBF (Mean ± SD) | Antibiofilm activity (%) | SBF (Mean ± SD) | Antibiofilm activity (%) |
| Blank (Media only) | 0.002 ± 0.002 | – | 0.002 ± 0.002 | |
| Control (Media + Bacteria) | 1.338 ± 0.003 | – | 1.430 ± 0.006 | |
| 75% of MBC | 0.126 ± 0.004 | 90.7 | 0.071 ± 0.004 | 95.1 |
| 50% of MBC | 0.188 ± 0.002 | 86.0 | 0.128 ± 0.003 | 91.2 |
| 25% of MBC | 0.250 ± 0.002 | 81.5 | 0.216 ± 0.004 | 85.0 |

1,5-Benzothiazepines derivative, against *S. epidermidis*, and moderate activity against *S. aureus*. To determine how Diltiazem HCl affects the *S. aureus* and *S. epidermidis* growth, the reduction of their ability to form biofilms was carried out as illustrated in the following section.

## Antibiofilm formation

Bacterial biofilms play an important role in the protection of pathogenic bacteria from conventional antibiotics (*Salem et al., 2022*). It is the root reason behind delayed growth, nutritional constraint, poor antibiotic penetration, adaptive stress responses, and the development of persistent cells that serve as a multi-layered defense against antibiotics (*Salem et al., 2022*). Disrupting biofilm formation may enhance the efficacy of antibacterial medications to clear infections involving biofilms that are refractory to current therapies (*Stewart, 2002*). Determining Diltiazem HCl's capacity to prevent *S. aureus* and *S. epidermidis* from forming biofilms was thus carried out.

The present findings showed that at 75%, 50%, and 25% of MBC (50 mg mL$^{-1}$), Diltiazem HCl decreased the formation of *S. aureus* biofilm by 90.7%, 86.0%, and 81.5%, respectively (Table 2). Furthermore, it decreased the production of *S. epidermidis* biofilm by 95.1%, 91.2%, and 85.0% at 75%, 50%, and 25% of its MBC (6.25 mg mL$^{-1}$), respectively (Table 2). The categorization of bacteria according to their specific biofilm formation's (SBF) ability was published by *Mittal et al. (2010)*. The groups included weak biofilm producers (SBF index < 1.00), intermediate biofilm producers (SBF index < 2.00), and strong biofilm producers (SBF index > 2.00). Accordingly, the ability of Diltiazem HCl to suppress the formation of both bacteria's biofilms was demonstrated by the significant reduction in SBF indices of *S. aureus* and *S. epidermidis* from SBF 1.338 and 1.433, respectively, to SBF indices <1.0 at varied concentrations (Table 2).

The drugs that impact calcium binding may also affect biofilm development because divalent cations, in particular calcium, are crucial bridging ions for bacterial polysaccharides, and biofilm formation, and they also play regulatory functions in bacterial gene expression (*Sarkisova et al., 2005*). While Diltiazem HCl was shown in the current study to be effective in decreasing the formation of biofilms in tested Gram-positive bacteria, specifically *S. aureus and S. epidermidis*, it was found to have the opposite effect on the inhibition of biofilms in Gram-negative bacteria, *Pseudomonas aeruginosa* as

reported by *Elkhatib, Haynes & Noreddin (2009)*. Its antagonist effect on Gram-negative bacteria was in contrast to other calcium channel blockers which showed an agonistic effect in the presence of conventional antibiotics (*Elkhatib, Haynes & Noreddin, 2009*). It seems that the specific medicine employed and the bacterial Gram stain types influence how different calcium channel blockers affect the production of biofilms (*Elkhatib, Haynes & Noreddin, 2009*). Further research is recommended to explore whether Diltiazem HCl's biofilm suppression of *S. aureus* and *S. epidermidis* works synergistically or antagonistically with conventional antibiotics.

## Antiviral activity

Despite the efficacy of conventional antivirals, treatment failures brought on by antiviral resistance are on the rise (*Abdelkarem et al., 2022*). Drug repurposing is a useful tool for investigating the antiviral potential of "non-antibiotic" drugs (*Singh et al., 2020*). The efficacy of 1,5-benzothiazepine moiety as the primary nucleus for antiviral derivatives was demonstrated (*Li et al., 2017*). Therefore, by using the MTT assay procedure, the antiviral activity of Diltiazem HCl was tested against the CoxB4 viruses compared to Acyclovir, a standard antiviral drug.

## Cytotoxicity against Vero cells

At first, the cytotoxicity of Diltiazem HCl and Acyclovir on Vero cells was assessed. The data analysis revealed dose-dependent effects, with higher cytotoxicity observed at higher drug concentrations and decreased cytotoxicity at lower concentrations (Fig. 2). To ensure accurate antiviral screening, non-cytotoxic doses were selected, and the half-maximal cytotoxic concentrations ($CC_{50}$) of Diltiazem HCl and Acyclovir, were assessed on Vero cells using the MTT assay (Table 3). Diltiazem HCl showed a $CC_{50}$ value of $310.37 \pm 0.067$ μg mL$^{-1}$ while Acyclovir exhibited a $CC_{50}$ value of $320.21 \pm 0.021$ μg mL$^{-1}$ which indicated that Diltiazem HCl has less toxicity than Acyclovir against the Vero cell line. The MNTC on Vero cells was 62.5 μg mL$^{-1}$ for both drugs; this concentration was used later for testing the antiviral activities.

## The morphological traits of Vero cells treated with various Diltiazem HCl concentrations

The results of this study showed that following Diltiazem HCl treatment, Vero cell morphology altered in a concentration-dependent way, resulting in either whole or partial monolayer loss (Fig. 3). In comparison to untreated cells, higher doses caused morphological abnormalities for some treated cells such as rounding, detachment, shrinking, and granulation. Lowering the concentrations below 62.5 μg mL$^{-1}$ resulted in a decrease in or elimination of these morphological changes without endangering the cell monolayer (Fig. 3).

## Assessment of the cell viability of the MNTC of Diltiazem HCl

The cell viability of control Vero cells and CoxB4-infected Vero cells were evaluated using an MTT assay. Vero cells were infected with the CoxB4 virus, and treated with and without Diltiazem HCl or Acyclovir for 48 h. The results showed that 99.99% of the control Vero

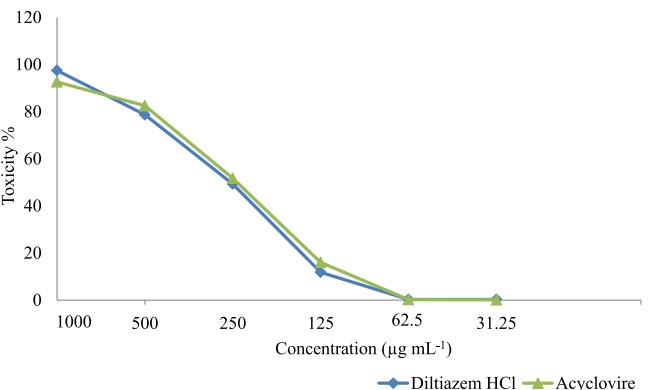

**Figure 2 Cytotoxicity of Diltiazem HCl and Acyclovir on Vero cell line at different concentrations.**

**Table 3 Antiviral activity of Diltiazem HCL against CoxB4 virus compared to the standard Acyclovir.**

| Drugs | MNTC | $CC_{50}$ | Antiviral effect (%) | $IC_{50}$ µg mL$^{-1}$ (mean ± SE) | SI |
|---|---|---|---|---|---|
| Diltiazem HCl | 62.5 | 310.37 ± 0.067 | 89.46 | 35.8 ± 0.54 | 8.66 |
| Acyclovir | 62.5 | 320.21 ± 0.021 | 77.71 | 42.71 ± 0.43 | 7.49 |

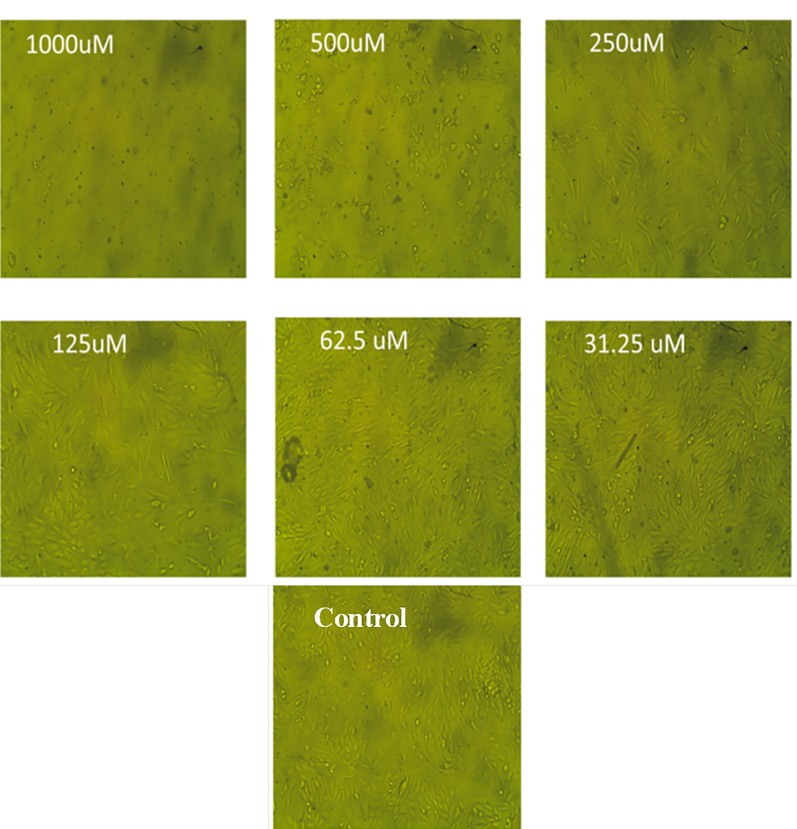

**Figure 3 The morphological characteristics of Vero cells treated with different concentrations of Diltiazem HCl.**
**Table 4 The viability (%) of CoxB4-infected Vero Cells treated with MNTC of Diltiazem HCL in comparison to the standard Acyclovir.**

| Tested cells | Control vero cells | CoxB4-infected cells | CoxB4-infected cells treated with Acyclovir | CoxB4-infected cells treated with Diltiazem HCl |
|---|---|---|---|---|
| Cell viability (%) | 99.99 | 40.44 | 81.35 | 93.72 |

cells were viable. However, Vero cell viability dramatically dropped to 40.44% in the presence of the CoxB4. The cell viability percent of CoxB4-infected cells treated with 62.55 µg mL$^{-1}$ of Acyclovir was 81.35%, and that of Diltiazem HCl was 93.72% (Table 4) indicating a markedly higher rate of cell survival compared to the untreated CoxB4-infected cells. These findings show that Diltiazem HCl administration partially restored the cell viability that had been considerably reduced by CoxB4 infection, suggesting that the drug has potential as an effective therapeutic agent for treating CoxB4 infection.

## Antiviral activity of Diltiazem HCl using the MTT assay technique

The reported bioactivities of Diltiazem HCl against Alzheimer's disease, influenza infection, and antitumor effects, make it an excellent model for drug repurposing against other infectious diseases such as CoxB4 infection (*Alluri et al., 2023*; *Kaur et al., 2023*; *Ribeiro et al., 2023*). Using the MTT assay, the antiviral activity of Diltiazem HCl against the CoxB4 virus was assessed at nontoxic doses (<62.5 µg mL$^{-1}$). It showed a significant antiviral efficacy against the CoxB4 viruses where its IC$_{50}$ (35.8 ± 0.54 µg mL$^{-1}$) was lower than that of the reference antiviral, Acyclovir (IC$_{50}$ 42.71 ± 0.43 µg mL$^{-1}$). The selectivity indices (SIs) of Diltiazem HCl and Acyclovir against the CoxB4 were calculated. The SIs were found to be 8.66 and 7.49 for Diltiazem HCl and Acyclovir, respectively, suggesting the potential of Diltiazem HCl as a promising antiviral agent against CoxB4.

Considering the role of calcium ion in the replication and morphogenesis of numerous viruses, these findings are corroborated by earlier research showing Diltiazem HCl's antiviral effectiveness against rotaviruses (*Khales et al., 2023*), rhinoviruses (*Gazina et al., 2005*), and influenza viruses (*Fujioka et al., 2018*). Also, Diltiazem did not exhibit antiviral activity against SARS-CoV-2; however, it strongly synergized the activity of the conventional drug Remdesivir (*Pizzorno et al., 2020*). It is worth noting that, not all Ca$_2^+$ channel inhibitors have appreciable antiviral activity (*Pizzorno et al., 2019*). Therefore, the observed antiviral activity of Diltiazem HCl may not be entirely explained by its induced modification of Ca$_2^+$ channel activity (*Pizzorno et al., 2020*). According to Fujioka's study, Diltiazem's antiviral effects are mostly due to its ability to control the expression of particular genes linked to cholesterol metabolism, the host antiviral response, and the induction of interferon-antiviral responses (*Fujioka et al., 2018*). The current study suggests that Diltiazem HCl could potentially be used to treat the CoxB4 virus infection, which would provide it an additional off-target effect.

## CONCLUSIONS

Drug repurposing represents a cutting-edge technique for the rapid and cost-effective identification of novel therapeutic applications for existing drugs. This study focused on investigating the antimicrobial properties of the antihypertensive Diltiazem HCl.
The results demonstrated that Diltiazem HCl exhibited significant antibacterial activity against Gram-positive pathogens, inhibiting their ability to form resistant biofilms. Furthermore, it displayed potent anti-CoxB4 properties, surpassing those of the standard Acyclovir. Consequently, Diltiazem HCl may be utilized to treat microbial infections, particularly those caused by the drug-resistant CoxB4 virus, *S. aureus*, and *S. epidermidis*. The current study provides a valuable off-target use of Diltiazem HCl as an antimicrobial agent.

## ABBREVIATIONS

| | |
|---|---|
| **CoxB4** | Coxsackie B4 virus |
| $\mathbf{Ca_2^+}$ | calcium |
| **MBC** | minimum bactericidal concentration |
| **SBF** | specific biofilm formation |
| **OD** | optical density |
| **MNTC** | maximum non-toxic concentration |
| **VERO** | adherent kidney epithelial cells from Cercopithecus aethiops |
| $\mathbf{CC_{50}}$ | half-maximal inhibitory concentration |
| **SI** | selectivity index |
| **MIC** | minimum inhibitory concentration |

## ACKNOWLEDGEMENTS

We are thankful to Dr. Ahmed M. A. A. Ramadan (Department of Botany and Microbiology, Faculty of Science, Al-Azhar University, Cairo, Egypt) for his kind help during this work.

### Funding

This work was supported and funded by the Deanship of Scientific Research at Imam Mohammad Ibn Saud Islamic University (IMSIU) (grant number IMSIU-RPP2023086). The funders had no role in study design, data collection and analysis, decision to publish, or preparation of the manuscript.

### Grant Disclosures

The following grant information was disclosed by the authors:
Deanship of Scientific Research at Imam Mohammad Ibn Saud Islamic University (IMSIU): IMSIU-RPP2023086.

## Competing Interests
The authors declare that they have no competing interests.

## Author Contributions

- Omar K. Alduaij conceived and designed the experiments, prepared figures and/or tables, and approved the final draft.
- Rageh K. Hussein analyzed the data, prepared figures and/or tables, authored or reviewed drafts of the article, and approved the final draft.
- Sharif Abu Alrub analyzed the data, authored or reviewed drafts of the article, and approved the final draft.
- Sabry A. H. Zidan conceived and designed the experiments, performed the experiments, prepared figures and/or tables, authored or reviewed drafts of the article, and approved the final draft.

## Data Availability
The raw measurements are available in the Supplemental File.

## Supplemental Information
Supplemental information for this article can be found online at http://dx.doi.org/10.7717/peerj.17809#supplemental-information.

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
