# Peer review of "Antimicrobial activities of Diltiazem Hydrochloride: drug repurposing approach"

_PeerJ, doi:10.7717/peerj.17809_

## Round 0.1 · original submission · Major Revisions

Dear Authors
Please address the issues raised by all the three reviewers and submit a revised version of your MS for re-consideration.

Reviewer 1 ·

Basic reporting

Please see report attached

Experimental design

Please see report attached

Validity of the findings

Please see report attached

Additional comments

Please see report attached

Annotated reviews are not available for download in order to protect the identity of reviewers who chose to remain anonymous.

Reviewer 2 ·

Basic reporting

The manuscript entitled "Antibiotic Activities of Diltiazem Hydrochloride: Drug Repurposing Approach" is well written and reports good results, drug repurposing could be a useful strategy to overcome the emerging resistance problem in the pathogenic microbes.

Minor revisions
1. Line 32-33: The terms Staff epidermidis, and Staff aureus are invalid, it could be Staph. but better use complete genus name or the established abbreviation (S) for these microbial names.
2. Line 79-80: Candida is yeast so it could be better to term it yeast rather than fungi, however in general its acceptable.
3. The % inhibition of biofilm can be termed as "Biofilm dispersion assay"
4. There is an extensive description of MTT assay in methodology, it should be rationalized with study design, and the procedure for antiviral assay should be concise and to the point
5. Figure 3 has low visibility, and needs improvement if included in the final manuscript

Experimental design

Experimental design of the study is good and fulfill the requirement of the research question, unnecessary details of the procedures should be avoided to make the manuscript concise and readable.

Validity of the findings

no comment

Additional comments

Although the study design is good and the results data sufficiently provide the evidence of the additional bio activities of the compound under investigation "Diltiazem Hydrochloride" against Gram positive bacteria and antiviral activity. However what is the take of authors on the suitability of the actual use of this drug as anti infectious compound, because the direct administration of the drug will definitely interfere with the blood pressure of the patient and especially those non hypertensives will be effected by it, so the practical use of the drug in patients as anti-infective agent is difficult, this point must be mentioned and discussed in discussion section of the manuscript. Moreover such additional bio-activities and effects are already reported in drug description by the manufacturers, this raise concerns on the novelty of the study and practicality of drug re-purposing. Authors need to clearly discuss these points in discussion section of the manuscript.

Reviewer 3 ·

Basic reporting

The language and expression need to be approved in many places in the manuscript. The author should avoid the use of terms such as our medication, our results, etc while describing the results. (please check the comment on the attached MS file).
In introduction section authors have added figure for drug structure without any citation that should be avoided.

Figures are not prepared well and are of low quality. The Ligand position and labeling of X-axis for each figure are not uniform. For example, figure 5.
There is no blank shown in figure 3.

Authors have used Staff epidermidis terminology for Staphylococcus epidermidis, which is not a proper scientific name.

Experimental design

The authors have to choose only E. coli, and three different strains of Staphylococcus other bacterial such as Pseduomonas etc are not selected. Similarly, to determine the antifungal effect they have used only 3 different strains of Candida sp. and no other fungal species have been testes. Selection of microbes is not appropriate. Methods in some sections are not clearly stated and are ambigous.
The equipment used for anaylsis are not mentioned throughout the manuscript. (Please check comments on attached file).

Validity of the findings

The authors have mentioned strong antibacterial efficacy however, this is not a strong efficacy, if compared with the control.
Figure 2 B (C. glabrata), shows a much higher zone for the standard antibiotic used but in the table 1 zone for 50 mg/ml MIC is given 12 ± 1, but it does not match with the figure 2B. The zone of inhibition is much smaller and negligible compared to the standard antibiotic. Thus, indicating contradictory results.

Similarly, the description of the results does not match. For example in Table 1, for S. epidermidis value is 20+3 at 50 mg/ml conc and 10± 2 at 6.25 mg/ml. however, in text lines 174-175 authors describe It recorded the highest zone of inhibition against S. epidermidis (20 ± 0.8 mm) with a minimum inhibitory concentration (MIC) 6.25 mg mL-1.

supplementary material shared is not data obtained from plate reader using software but is manually added values in Excel files shared.

Annotated reviews are not available for download in order to protect the identity of reviewers who chose to remain anonymous.

---

## Round 0.2 · Minor Revisions

Dear Authors
Thank you for revising the manuscript as per review reports. Although, you have addressed most of the queries raised by the reviewers but one of the reviewers has raised some minor issues in the MS. I request you to please address all those issues carefully and resubmit the MS.

Reviewer 1 ·

Basic reporting

The author has incorporated the changes

Experimental design

The author has incorporated the changes

Validity of the findings

The author has incorporated the changes

Additional comments

The author has incorporated the changes

Reviewer 2 ·

Basic reporting

The authors have incorporated the recommended revisions and the manuscript is updated, it cab accepted.

Experimental design

Satisfactory

Validity of the findings

Satisfactory

Reviewer 3 ·

Basic reporting

Review and Suggestions for Improvement

1. Language
The authors have improved the language throughout the manuscript, but there are still many grammatical and punctuation errors, as well as missing articles (e.g., "the"). Thus, further improvements are needed for grammar and clarity. Below are specific examples and their corrections:

Line 25
- Original: "This study's primary objective was..."
- Corrected: "The primary objective of this study was to evaluate..."

Lines 33-34
- Original: "of Diltiazem HCl against CoxB4 virus"
- Corrected: "of Diltiazem HCl against the CoxB4 virus"

Line 39:
- Original: "be potent against CoxB4 virus."
- Corrected: "be potent against the CoxB4 virus."

Line 110:
- Original: "supernatant was dispose of and every well"
- Corrected: "supernatant was disposed of and every well"

Line 113:
- Original: "Finally, to solubilize the dye; 250 μL of 95%..."
- Corrected: "Finally, to solubilize the dye, 250 μL of 95%..."

Line 131:
- Original: "Dulbecco’s phosphate buffered saline medium., and 0.1 mL of each..."
- Corrected: "Dulbecco’s phosphate-buffered saline medium, and 0.1 mL of each..."

Similarly, in methods, for a single medium, media term is used at many places.

Such mistakes are present throughout the manuscript. Many such mistakes are marked with comments on the file shared. However, the authors are advised to revise the article carefully.

2. Literature References, and Sufficient Field Background/Context Provided
In the manuscript, the authors have cited relevant literature and discussed the appropriate background. However, it would be better to add literature support from 2024 to show the current research scenario.

3. Professional Article Structure, Figures, Tables. Raw Data Shared
Figure 1:
The professional article structure is satisfactory. In the previous version, the authors included a general figure: "Chemical structures of 1, 5-benzothiazepine moiety (A), and its derivatives Diltiazem HCl (B)" in the introduction. Now they have moved Figure 1 to the results and discussion section. This figure is neither generated by the authors nor obtained from NMR data by the authors. Thus, it is not part of their results. Furthermore, the figure is of low resolution. It is recommended to remove this figure from the manuscript or use it for a graphical abstract after quality improvement.

Figure 2F (S. aureus):
In Figure 2, there is no zone of inhibition shown at any concentration. This does not match the results description (i.e., 12 ± 2 mm, and the MIC at 50 mg/mL). Authors need to add an image of the plate showing a clear zone of inhibition to support their claim.

Figure 3:
Figure 3 represents the effect of Diltiazem concentration on SBF by S. aureus and S. epidermidis at different MCBs. The same data is also presented in Table 2. Presenting the same data in both a figure and a table is redundant. It is recommended to present the data in only one format. Furthermore, the quality and resolution of the figure need improvement.

Figure 4:
The quality and resolution of Figure 4 need improvement. The labeling of the axes should be centered (below for the x-axis and on the side of the y-axis). The position of ligands also needs to be improved.

Tables:
All the tables should have captions starting with "Table 1." Need corrections
The title of Table 1 needs improvement for clarity and structure.

Experimental design

Experimental design is satisfactory .
However, upon reviewing the manuscript, it was noted that the concentration of streptomycin mentioned in the methodology section is 100 g/mL (line 123), which is excessively high. Typically, streptomycin is used at concentrations around µg or mg/mL not grams per milliliter. This appears to be a typographical error. Please verify it. If it was the actual concentration used. What was the reference?

"On line 95-96: "..the bacterial and fungal spore suspensions were equally spread on sterile Petri plates."
The bacterial species used in this research are non-spore formers. Therefore, it is important to clarify how bacterial spores were involved in this case. It should be noted that only Serratia marcescens subspecies sakuensis has been reported to form spores. Please correct this error."

Validity of the findings

There is need to improve figures but the conclusion is well stated and linked to the research

Annotated reviews are not available for download in order to protect the identity of reviewers who chose to remain anonymous.

---

## Round 0.3 · accepted · Accept

In the second round of review, one reviewer raised some minor queries which have been addressed by the authors, therefore the manuscript is accepted for publication.